# Validamycin Inhibits the Synthesis and Metabolism of Trehalose and Chitin in the Oriental Fruit Fly, *Bactrocera dorsalis* (Hendel)

**DOI:** 10.3390/insects14080671

**Published:** 2023-07-28

**Authors:** Ying Li, Yonghong Xu, Shunjiao Wu, Baohe Wang, Yaying Li, Yinghong Liu, Jia Wang

**Affiliations:** 1Key Laboratory of Agricultural Biosafety and Green Production of Upper Yangtze River (Ministry of Education), Southwest University, Chongqing 400715, China; liying6611@163.com (Y.L.); xyh202202@126.com (Y.X.); wushunjiao0130@126.com (S.W.); wangbh0424@163.com (B.W.); zliyaying@swu.edu.cn (Y.L.); 2College of Plant Protection, Southwest University, Chongqing 400715, China

**Keywords:** *Bactrocera dorsalis*, validamycin, trehalose, trehalase, trehalose-6-phosphate phosphatase, chitin, chitinase, imaginal disc growth factor

## Abstract

**Simple Summary:**

Validamycin is a strong competitive trehalase inhibitor that has long been used as an anti-fungal antibiotic to target trehalose hydrolysis. To clarify whether validamycin can also be adopted for the management of *Bactrocera dorsalis* (Diptera, Tephritidae), the effects of validamycin injection on the synthesis and metabolism of trehalose and chitin were evaluated. The results showed that validamycin injection significantly inhibited the synthesis and metabolism of trehalose and chitin, thus leading to high mortality and deformity rates. These findings imply that validamycin can be considered as a promising potential insecticide for the management of *B. dorsalis*.

**Abstract:**

The oriental fruit fly, *Bactrocera dorsalis* (Hendel), is a notorious invasive pest that has raised concerns worldwide. Validamycin has been demonstrated to be a very strong inhibitor against trehalase in a variety of organisms. However, whether validamycin can inhibit trehalase activity to suppress trehalose hydrolysis and affect any other relevant physiological pathways in *B. dorsalis* remains unknown. In this study, the effects of validamycin injection on the synthesis and metabolism of trehalose and chitin were evaluated. The results show that validamycin injection significantly affected trehalase activity and caused trehalose accumulation. In addition, the downstream pathways of trehalose hydrolysis, including the synthesis and metabolism of chitin, were also remarkably affected as the expressions of the key genes in these pathways were significantly regulated and the chitin contents were changed accordingly. Intriguingly, the upstream trehalose synthesis was also affected by validamycin injection due to the variations in the expression levels of key genes, especially *BdTPPC1*. Moreover, BdTPPC1 was predicted to have a binding affinity to validamycin, and the subsequent in vitro recombinant enzyme activity assay verified the inhibitory effect of validamycin on BdTPPC1 activity for the first time. These findings collectively indicate that validamycin can be considered as a promising potential insecticide for the management of *B. dorsalis*.

## 1. Introduction

Trehalose, the main sugar circulating in the insect haemolymph [1,2], plays important roles in diverse physiological processes, such as being used as an source of energy to meet the demand of insect development and metamorphosis [3], participating in the resistance to abiotic stresses [2,4], and serving as a starting material for downstream chitin synthesis [1,2,5]. There are five pathways of trehalose synthesis in organisms [6,7], including maltooligosyl trehalose synthase (TreY)/maltooligosyl trehalose trehalohyorotase (TreZ), trehalose synthase (TreS), trehalose phosphorylase (TreP), trehalose glycosyltransferring synthase (TreT), and trehalose-6-phosphate phosphatase (TPP)/trehalose-6-phosphate synthase (TPS). The TPS−TPP pathway is the dominated one in insects [2,8]. Trehalase (Tre) is currently the only known enzyme that can irreversibly hydrolyze trehalose in organisms, and has been demonstrated to be the first enzyme in chitin synthesis pathway [9].

Chitin is the main component of the cuticle and peritrophic membranes of insects [10,11,12,13,14], and the disruption of the chitin metabolism has a lethal effect on insects [15,16,17,18,19]. For example, the silencing a chitin synthase of *Spodoptera frugiperda* (Lepidoptera: Noctuidae) that is highly expressed in the cuticle caused a high rate of larval mortality and pupal malformation [20]. Additionally, inhibiting chitinase 10 (*Cht10*) of *Aedes albopictus* (Diptera: Culicidae) expression led to abnormal molting, increased mortality; decreased chitin content; and thinning of the pupal epiculticle, endocuticle, and midgut wall [21]. Therefore, the chitin biosynthesis pathway has been recognized as a promising target for pest management [22,23]. Among the key genes in the chitin synthesis pathway is Tre, which has raised plenty of interest and several insecticidal methods targeting Tre have been developed, such as delivering dsRNA [24] and environmentally friendly compounds [25]. Moreover, the absence of trehalose in mammals confers much more safety on Tre-targeting pest management methods [26,27].

For decades, validamycin, an analogue of trehalose, has been widely used as an anti-fungal agricultural antibiotic to control plant diseases, such as rice sheath blight [28,29]. As Tre is an indispensable enzyme in insects and validamycin can bind tightly to Tre, it had long been considered as a promising competitive Tre inhibitor to manage insects as well [25]. Indeed, this inhibitor has been demonstrated to be able to affect mobility, feeding, metabolism, growth, development, reproduction, and flight of insects. For instance, inhibiting Tre activity by validamycin reduced the expression levels of chitin metabolism-related genes and led to a malformed *Diaphorina citri* (Hemiptera, Psyllidae) phenotype [30]. Moreover, silencing of the chitinase-like protein *ENO3* and *CP7* genes in *Diaphorina citri* by RNAi resulted in abnormal phenotypes [31].

The oriental fruit fly (*Bactrocera dorsalis*) is an important agricultural pest in Asia, Africa, and the Pacific region, affecting over 150 types of fruits and vegetables in tropical and subtropical areas [32,33]. *B. dorsalis* causes damage by larvae feeding within the fruits and vegetables [34]. For the past few decades, the management of oriental fruit flies and several other economic pests has relied heavily on the application of chemical pesticides due to their effectiveness and quick response [35]. However, the extensive and unmonitored use of chemical pesticides in orchards can potentially cause various insect pests on non-target organisms and the environment, as well as lead to the development of resistance against different insecticides [36,37]. Therefore, formulating environmentally friendly and sustainable pest control strategies is an urgent need. Although validamycin has detrimental effects on several insects, whether it can be used as an insecticide for managing *B. dorsalis* remains unknown and necessitates further investigation. In this endeavor, the effects of validamycin treatment on the trehalose and chitin metabolism, as well as the performance of *B. dorsalis,* were explored in the present study. These findings implied that validamycin is a promising potential insecticide and laid the basis for further developing a methodology for utilizing validamycin in *B. dorsalis* management.

## 2. Materials and Methods

### 2.1. Insects

The pupae of *B. dorsalis* were collected from a citrus orchard in Chengjiang (Yunnan, China) in 2021. All of the insects were reared at 27 ± 1 °C, 75% relative humidity, and over a 14 L:10 D photoperiod. The larvae were fed with an artificial diet composed of yeast powder, sugar, wheat germ powder, and corn meal, while the adult insects were fed with an artificial diet composed of yeast powder, sugar, honey, and water [38]. In this study, the individuals used for the experiments were reared under the same conditions.

### 2.2. Injection of Validamycin

The validamycin (Huaxia Chemical Reagent, Chengdu, China) was dissolved in ddH_2_O to obtain 5 μg/μL and 10 μg/μL solutions. The third-instar larvae (about 4 mm in length) were collected and each individual was injected with 400 nL validamycin solution using a Nanoject II Auto-Nanoliter Injector (Drummond Scientific, Broomall, PA, USA) [39]. Larvae injected with 400 nL ddH_2_O were taken as the control. The phenotype of individuals in each treatment was recorded daily until pupation. The representative images of individuals from each treatment were acquired using a DFC digital microscope camera attached to a Leica M205A (Danaher, Wetzlar, Germany). Three replicates (*n* = 200) were set for each treatment. After injection, the dead larvae in each treatment were removed daily and the remaining individuals were used for the subsequent experiments.

### 2.3. Measurement of Trehalase Activity

Then, 24 and 48 h after injection, 10 individuals were randomly selected from each treatment to measure the trehalase activities. Briefly, the samples were homogenated in 1mL extraction solution (Solarbio, Beijing, China) and centrifuged at 8000× *g* for 20 min at 4 °C. The supernatant (350 μL) was centrifuged again at 20,800× *g* for 10 min at 4 °C to remove debris. An aliquot of supernatant was used for protein content determination using the BCA Protein Assay Kit (Beyotime, Shanghai, China). The other aliquots of supernatant were used for measuring the trehalase activities using the Trehalase Activity Assay Kit (Solarbio, Beijing, China), according to the manufacturer’s instructions. Three replicates were set for each treatment.

### 2.4. Measurement of Carbohydrate Content

Then, 24 and 48 h after injection, 5 individuals were collected for measuring the trehalose, glycogen, and glucose content using the Trehalose Assay Kit (Solarbio), Glycogen Assay Kit (Solarbio), and Glucose Assay Kit (Solarbio), respectively, following the manufacturer’s instructions. Three replicates were set for each treatment.

### 2.5. Measurement of Chitin Content

Then, 24 and 48 h after injection, 10 individuals were collected and homogenated in 1 mL PBS (pH 7.2–7.4). After centrifugation at 12,000× *g* for 5 min, the supernatant was collected for chitin measurement. The chitin contents were measured using the Insect chitin enzyme-linked immunosorbent assay kit (YaJi Biological, Wuhan, China), according to the manufacturer’s protocols. Three replicates were set for each treatment.

### 2.6. Quantitative Real-Time PCR (qRT-PCR)

The total RNA was extracted from the larvae 48 h after injection, using TRIzol reagent (Invitrogen Life Technologies, Carlsbad, CA, USA) following the manufacturer’s instructions. The RNA concentrations were determined using a Nanovue UV−VIS spectrophotometer (GE Healthcare, Fairfield, CT, USA). The cDNA was synthesized using an Evo M-MLV Mix Kit with gDNA clean for qPCR (Accurate Biotechnology, Changsha, China). The expression levels of the genes related to the trehalose metabolism and chitin synthesis pathways [8,19] were detected by qRT-PCR with specific primers (Appendix A) and *α-tubulin* was used as a reference gene. qRT-PCR was conducted using the Bio-Rad CFX96^TM^ Real-Time PCR Detection System (Bio-Rad, Hercules, CA, USA) in 10 μL reaction volume, including 5 μL SYBR^®^ Premix Ex Taq II (Takara, Shiga, Japan), 0.5 μL forward and reverse primers, 3.5 μL ddH_2_O, and 0.5 μL cDNA template. The reaction conditions were as follows: 94 °C for 3 min, followed by 35 cycles of 94 °C for 10 s, and 60 °C for 30 s. The relative gene expression levels were calculated using the 2^−ΔΔCT^ method [40]. Four biological and three technical replicates were set for each gene.

### 2.7. Modelling of Trehalose-6-Phosphate Phosphatase C1 (BdTPPC1) and BdTPPC1−Validamycin Docking

The tridimensional structures of validamycin were obtained from the PubChem Compound Database and converted to pdb format [41]. The deduced amino acid sequences of trehalose-6-phosphate phosphatase (BdTPPC1) were submitted to the Phyre^2^ server (http://www.sbg.bio.ic.ac.uk/phyre2/html/page.cgi?id=index, accessed on 15 November 2022) for prediction of the tridimensional structures [42]. Then, the ligand and receptors were optimized using Autodock Vina software https://autodock.scripps.edu/, accessed on 20 July 2023 [43]. The molecular docking of validamycin to BdTPPC1 was modeled using the software PyPx 0.8. The most possible docking modes were further analyzed and visualized using the software Discovery Studio 2019 Client.

### 2.8. Heterologous Expression of Recombinant BdTPPC1

The nucleotide sequence of BdTPPC1 was synthesized by Zoonbio Biotechnology (Nanjing, China). The synthesized BdTPPC1 was double-digested with BamHI and HindIII, and the purified fragment was subcloned into the vector pFastBac1. Then, the plasmid was transfected with transfection reagent lipofectamine^®^ 2000 (Invitrogen, Carlsbad, CA, USA) into SF9 cells for recombinant protein expression. Western blot analysis was performed to confirm the expression of BdTPPC1. The recombinant protein was then purified using Ni column chromatography and stored at −80 °C for the subsequent experiments.

### 2.9. Inhibition of Recombinant BdTPPC1 Activity by Validamycin

The purified recombinant BdTPPC1 was diluted in Tris-Hcl solution (pH 8) to a concentration of 0.6 mM. The validamycin was dissolved in Tris-Hcl solution (pH 8) to different concentrations (0 mM, 5 mM, 10 mM, and 15 mM). The BdTPPC1 activities were measured following the method described by Klutts et al. [44] with minor modifications. Briefly, the first reaction system containing 10 μL BdTPPC1 solution, 20 μL validamycin solution of different concentrations, 30 mM Tris-HCl (pH 7.4), 2.5 mM MgCl_2_, 10 mM trehalose-6-phosphate, and ddH_2_O to 100 μL was incubated at 35 °C for 30 min. After incubation, 100 μL color development system, including 0.15% malachite green oxalate and 1% (NH_4_)_2_MoO_4_ dissolved in 38% HCl, was added and incubated at 35 °C for 2 min. The reaction systems without incubation were taken as the control. Then, we measured the absorbance at 630 nm using a spectrophotometer. The inorganic phosphate concentration was calculated based on the standard curve. The TPPC1 activity was expressed as μmol Pi/mg protein/min. Three biological and three technical replicates were set for each treatment.

### 2.10. Statistical Analyses

Statistics were performed using GraphPad Prism software https://www.graphpad.com/, accessed on 20 July 2023 (GraphPad Software, Santiago, CA, USA). One-way ANOVA and Tukey’s test were performed to evaluate the significance of differences among multiple groups after verification of the normality and homogeneity of variances.

## 3. Results

### 3.1. Inhibition of Trehalase Activity by Validamycin

The injection of validamycin significantly impacted the trehalose hydrolysis in *B. dorsalis* (Figure 1). Then, 24 and 48 h after injection, the Tre activities were decreased in validamycin-injected larvae compared with that of the control (Figure 1A). Accordingly, trehalose was accumulated (Figure 1B), while the glucose was reduced (Figure 1C) in the validamycin-injected larvae. Compared with the injection of 5 μg/μL validamycin, that of a higher dose exerted a stronger effect on trehalose hydrolysis (Figure 1).

### 3.2. Inhibition of Chitin Biosynthesis by Validamycin

The chitin biosynthesis pathway started with trehalose hydrolysis. Validamycin injection significantly inhibited chitin biosynthesis in a dose-dependent manner (Figure 2A). The expressions of most genes in the chitin biosynthesis pathway were down-regulated by both validamycin treatment, but showed marginal disparity between each other 48 h after injection (Figure 2C–J). Intriguingly, the expressions of *BdTre* were stimulated by validamycin, at a high concentration in particular (Figure 2B).

### 3.3. Effect of Validamycin Injection on the Expressions of Chitin-Degradation-Related Genes

The expression levels of seven chitinase genes (*Chts*) and five imaginal disc growth factor genes (*IDGFs*) in the *B. dorsalis* larvae were detected 48 h after validamycin injection (Figure 3). The expressions of *BdCht1* and *BdCht8* were suppressed in both validamycin treatments (Figure 3A,E). The expressions of *BdCht5* and *BdCht10* were only suppressed by 10 μg/μL validamycin (Figure 3C,F). In contrast, the expressions of *BdCht7* and *BdCht11* were stimulated by validamycin treatments (Figure 3D,G). The expressions of *BdCht2* did not show any significant change (Figure 3B). Except for *BdIDGF4* (Figure 3K), all *BdIDGFs* were up-regulated by validamycin treatments, especially 10 μg/μL validamycin (Figure 3H–J,L).

### 3.4. Effect of Validamycin Injection on the Expressions of Trehalose Biosynthesis Related Genes

The expressions of *BdTPS* and *BdTPPs* in the *B. dorsalis* larvae were detected 48 h after validamycin injection (Figure 4). The injection of 5 μg/μL validamycin up-regulated the expressions of *BdTPS* and *BdTPPB*, and the injection of 10 μg/μL validamycin did not significantly change the expression levels of *BdTPS* and *BdTPPB* (Figure 4A,B). The expressions of *BdTPPC1* were down-regulated in both validamycin treatments (Figure 4C). The expressions of *BdTPPC2* were not significantly changed (Figure 4D).

### 3.5. Modelling of BdTPPC1 and BdTPPC1−Validamycin Docking

BdTPPC1 possesses two distinct structural domains, a cap domain composed of four antiparallel β-strands and two α-helices, and a hydrolase domain featuring central eight-twisted β-strands and five α-helices. The docking simulation of BdTPPC1−validamycin illustrated that the substrate successfully entered the catalytic pocket formed at the interface between the cap and hydrolase domain, and the binding free energy was −6.4 kcal/mol (Figure 5A,B). The in vitro recombinant enzyme activity assay verified that the BdTPPC1 activities were inhibited by validamycin in a dose-dependent manner (Figure 5C).

### 3.6. Phenotypes of B. dorsalis after Validamycin Injection

The larval mortality and pupal deformity rates were significantly increased after validamycin injection, especially 10 μg/μL validamycin (Figure 6A,B). Unlike the normal pupae, the deformed pupae did not present an oval shape and their body segments were twisted (Figure 6C). The larvae injected with the effective validamycin died with their bodies an orange color, while the control group showed no change.

## 4. Discussion

Validamycin has been demonstrated to be a very strong inhibitor against trehalase in a variety of organisms, such as insects, nematodes, and fungi, and thus has raised plenty of interest in recent years [45,46,47]. As a matter of fact, validamycin has long been used as an anti-fungal agricultural antibiotic to control rice sheath blight due to its superior property of high efficiency and environmental friendliness [48]. Currently, although validamycin is not commonly used for managing insect pests, a large body of evidence has pointed out the potential feasibility in this field. For example, a toxicological test analysis revealed that *Spodoptera litura* (Lepidoptera: Noctuidae) larval growth and development was significantly inhibited and the pupation rate was significantly reduced after validamycin treatment [48]. In addition, validamycin treatment significantly increased the mortality rate of *Diaphorina citri* adult insects, reduced the pupation rate, and caused significant deformities in adult insects [30]. In this study, we found that the injection of validamycin into *B. dorsalis* larvae could remarkably impair trehalose hydrolysis (Figure 3), which is consistent with the findings of previous studies that the application of validamycin inhibited the conversion from trehalose to glucose [18,30,39,49,50], implying that the validamycin can also be considered as a potential insecticide for *B. dorsalis* management by targeting trehalose hydrolysis and downstream physiological processes.

Chitin, a major component of insect cuticles, tracheae, and peritrophic membranes [51], plays important roles in the growth and metamorphosis of insects. The chitin biosynthesis pathway starts with trehalose hydrolysis [52]. Previous studies have shown that the disruption of trehalose hydrolysis significantly affected downstream chitin biosynthesis and led to high mortality [53]. Likewise, the injection of validamycin into *B. dorsalis* larvae resulted in the suppression of chitin biosynthesis, as key genes in chitin biosynthesis were down-regulated and the chitin contents were reduced (Figure 2). Apart from chitin biosynthesis, chitin degradation also participates in many vital physiological processes of insects, such as molting [54]. *Chts* were found to be hydrolases that decompose chitin into N-acetylglucosamines [55]. In *B. dorsalis*, *BdChts* members play different roles, although they are all involved in chitin degradation [55,56,57]. *IDGFs* were certified as structural proteins to maintain the epithelial apical extracellular matrix (ECM) scaffold against chitinolytic degradation [58]. In this study, validamycin injection significantly affected the expressions of most *BdChts* and *BdIDGFs*, indicating that chitin degradation was also disrupted by validamycin. The severe impact of validamycin on chitin synthesis and metabolism may be one of the major reasons for the high mortality and deformity rates (Figure 6) [49].

Validamycin functions as a competitive Tre inhibitor because of its structural similarity with trehalose, the substrate of Tre catalyzation [59]. Additionally, validamycin has also been found to be able to inhibit TPS, possibly attributed to its structural similarity with trehalose-6-phosphate, the product of TPS catalyzation [60]. In insects, trehalose-6-phosphate serves as the substrate of TPP for trehalose formation; therefore, we presumed that validamycin could also affect TPP activity. Indeed, the prediction of the BdTPPC1 structure and BdTPPC1−validamycin docking illustrated that BdTPPC1 possesses an affinity to validamycin (Figure 5A). Furthermore, validamycin was demonstrated to be able to inhibit recombinant TPP activities in vitro in a dose-dependent manner for the first time (Figure 5C). After the injection of 5 μg/μL and 10 μg/μL of validamycin, both the glucose content and the expression levels of the genes related to the chitin synthesis pathway were significantly down-regulated. However, for *BdTPS* and *BdTPPB* expressions, injection of 5 μg/μL validamycin had a stimulating effect, possibly due to inhibition of the downstream trehalose hydrolysis and chitin biosynthesis pathway, which may trigger a feedback mechanism to increase trehalose synthesis. In previous studies, similar results were observed, where the expression level of *NlTPS* injecting 0.1 μg/μL of validamycin was significantly higher compared with injecting 10 μg/μL [39]. In addition, the expression level of *BdTPPC1* was remarkably suppressed after the injection of validamycin (Figure 4B). These findings collectively indicate that validamycin can inhibit both trehalose formation and hydrolysis. Nevertheless, its inhibitory effect on trehalose formation is possibly overwhelmed by the inhibitory effect on trehalose hydrolysis, leading to trehalose accumulation in validamycin-injected insects (Figure 1). Noteworthy, a lot of insect species lack TPPs, in that TPSs have incorporated an active TPP domain over a long history of evolution to become fuse proteins and function as both TPS and TPP [61]. However, dipteran insects, especially tephritidae species, possess abundant independent active TPPs [61]. We suppose that the other TPPs of *B. dorsalis* can also be affected by validamycin. If that is the case, the tephritidae species may harbor more validamycin targets than other insects, which could confer them with a higher sensitivity to this compound.

The findings in this study indicated that validamycin has the potential to manage *B. dorsalis* by interfering in the synthesis and metabolism of trehalose and chitin. However, it is obviously not feasible to deliver this compound by injection as has been done in this study. Additionally, larvae are not the ideal targets for validamycin treatment as there have been no reports indicating that validamycin can penetrate the fruit and kill the larvae feeding inside. Therefore, it is urgent to develop proper methods for the application of validamycin. In fact, the practical utilizations of validamycin for managing insect pests in the field are still rare, largely owing to the lack of ideal delivery systems. As only adults of *B. dorsalis* can be exposed directly to the exogenous compounds, it is supposed to be better to apply validamycin against adults. The reason we did not take adults as experimental subjects in this study is that we intended to verify the toxicity of validamycin to *B. dorsalis* first, and it was more convenient to take larvae as experimental subjects for a large number of injections. Nevertheless, the control efficiency of validamycin to adults of *B. dorsalis* needs to be further assessed. Moreover, given that the cuticle barrier of insects may attenuate the absorption of validamycin, it could possibly be better to deliver this compound by feeding, such as through poison bait. The preliminary investigation indicated that validamycin treatment on adults may affect their fecundity, and a thorough evaluation is still ongoing.

## 5. Conclusions

This study shows that the validamycin can significantly affect the synthesis and metabolism of trehalose and chitin in *B. dorsalis* larvae, thus leading to high mortality and deformity rates. These findings indicate that validamycin can be considered as a promising potential insecticide for the management of *B. dorsalis*.

## Figures and Tables

**Figure 1 insects-14-00671-f001:**
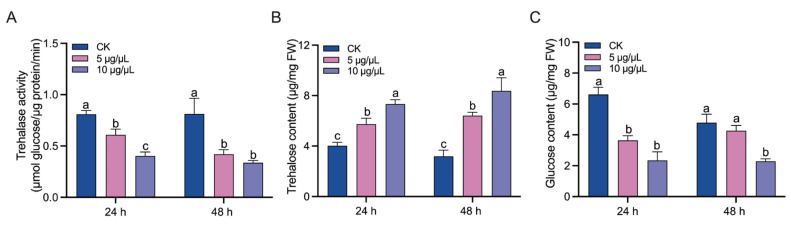
The effect of validamycin injection on trehalase activity in *Bactrocera dorsalis*: (**A**) trehalase activity, (**B**) trehalose content, and (**C**) glucose content. Error bars indicate mean ± SEM. Different letters above the bars indicate significant differences (*p* < 0.05).

**Figure 2 insects-14-00671-f002:**
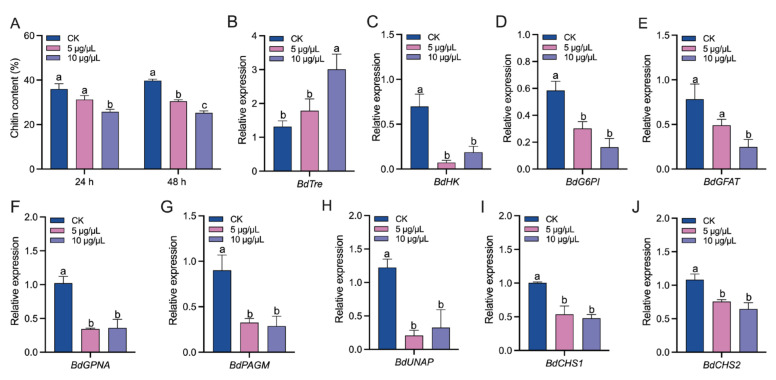
The effect of validamycin injection on chitin biosynthesis in *Bactrocera dorsalis*. (**A**) Chitin content. (**B**–**J**) Relative expression levels of genes in the chitin synthesis pathway. Error bars indicate mean ± SEM. Different letters above the error bars indicate significant differences (*p* < 0.05).

**Figure 3 insects-14-00671-f003:**
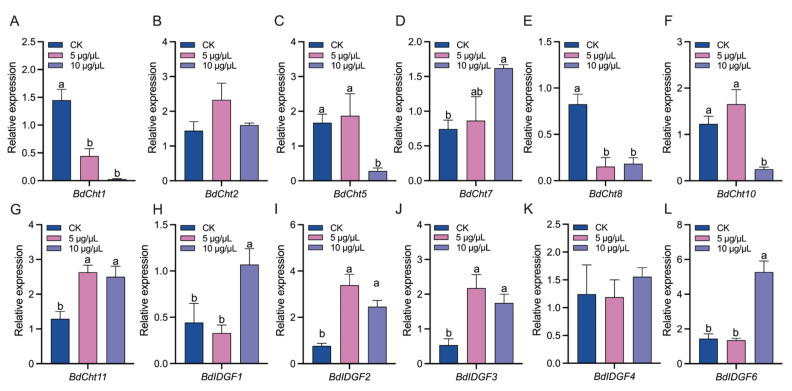
The effect of validamycin injection on the expressions of chitin-degradation-related genes in *Bactrocera dorsalis*. (**A**–**G**) Relative expression levels of *BdChts*. (**H**–**L**) Relative expression levels of *BdIDGFs*. Error bars indicate mean ± SEM. Different letters above the error bars indicate significant differences (*p* < 0.05).

**Figure 4 insects-14-00671-f004:**
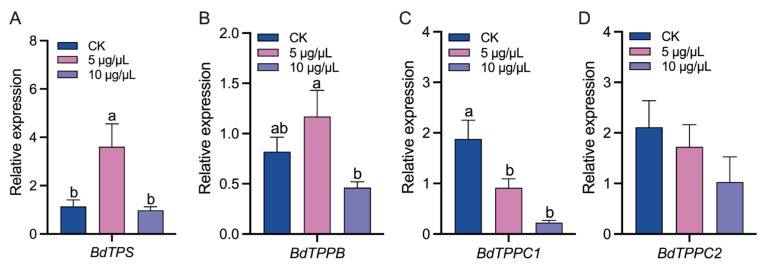
The effect of validamycin injection on the expressions of trehalose-biosynthesis-related genes in *Bactrocera dorsalis*. (**A**) The relative expression level of *BdTPS*. (**B**–**D**) The relative expression levels of *BdTPPs*. Error bars indicate mean ± SEM. Different letters above the error bars indicate significant differences (*p* < 0.05).

**Figure 5 insects-14-00671-f005:**
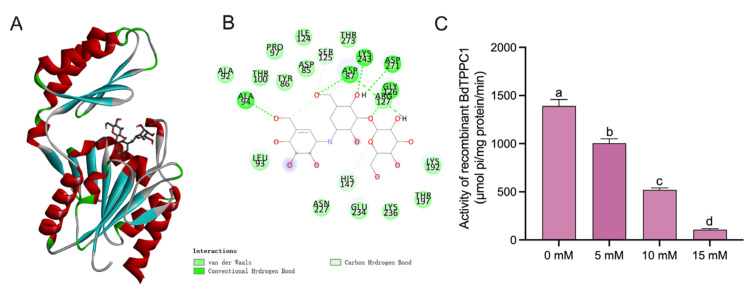
Inhibitory effect of validamycin on BdTPPC1 activity: (**A**) simulation of BdTPPC1−validamycin docking, (**B**) binding sites of validamycin to BdTPPC1, (**C**) recombinant BdTPPC1 activity treated with different concentrations of validamycin to BdTPPC1. Error bars indicate mean ± SEM. Different letters above the error bars indicate significant differences (*p* < 0.05).

**Figure 6 insects-14-00671-f006:**
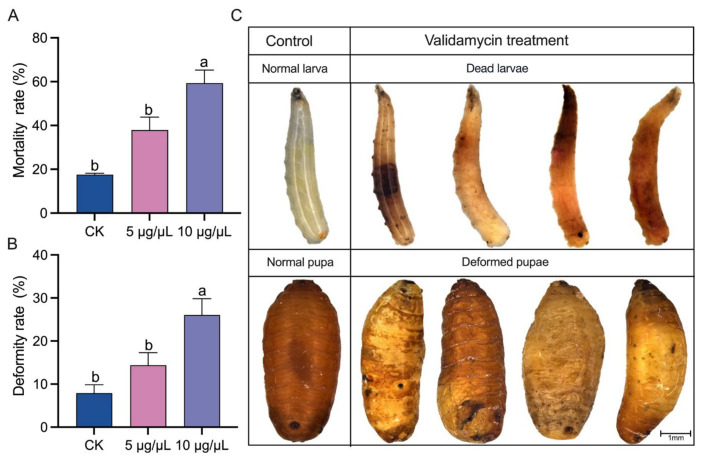
Phenotypes of *Bactrocera dorsalis* after validamycin injection: (**A**) mortality rate, (**B**) deformity rate, and (**C**) representative images of the phenotypes. Error bars indicate the mean ± SEM. Different letters above the error bars indicate significant differences (*p* < 0.05).

## Data Availability

The data presented in this study are available upon request from the corresponding author.

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
