# Peer review of "Validamycin Inhibits the Synthesis and Metabolism of Trehalose and Chitin in the Oriental Fruit Fly, Bactrocera dorsalis (Hendel)"

_insects, 2023, doi:10.3390/insects14080671_

Round 1

Reviewer 1 Report

This MS entitled “Validamycin Inhibits the Synthesis and Metabolism of Trehalose and chitin in the Oriental Fruit Fly, Bactrocera dorsalis (Hendel)” by Li and colleagues describe effects of validamycin injection into B. dorsalis on trehalose metabolism and chitin synthesis. Validamycin injection (2 or 4 ug/insect) induced mortality of larvae and malformation of pupae. The effects of validamycin on expressions of genes which are involved in trehalose metabolism and cuticle synthesis/metabolism pathways were assayed. The activity of recombinant trehalose-6-phosphate phosphatase (BdTPPC1) was inhibited by validamycin.  

The experiments have been designed well and the conclusions drawn are supported by data included in the MS. I have a few minor points suggested to change prior to publication.

1) Most of experiments were assayed with high dose validamycin (2 or 4 ug/insect).  Please show any references on what was the basis for assay at these concentrations.

2) Line 51, “thinning of the pupal epidermis,” is this sentence correct? In ref18, it is shown that thinning of the pupal epicuticle, not epidermis.

3) Line 158, “at 35 C”, it lacks “°”.

5) To clarify the effect of validamycin according to the duration of exposure, it is better that the rate of mortality and deformity are displayed on each day. 

6) Please mention more detail of dead larvae which were injected validamycin. Is that caused by defect of pupal cuticle or lack of enagy

7) In figure 6, the malformations were detected at pupation. Do you think the similar effect are observed on larval ecdysis if validamycin is injected at 2nd larval instar?

Author Response

Thank you for your letter and for the reviewers’ comments concerning our manuscript entitled “Validamycin Inhibits the Synthesis and Metabolism of Trehalose and Chitin in the Oriental Fruit Fly, Bactrocera dorsalis (Hendel)” (ID: insects-2498817). Those comments are all valuable and very helpful for revising and improving our paper, as well as the important guiding significance to our researches. We have studied comments carefully and have made correction which we hope meet with approval. Revised portion are marked in red in the paper. The main corrections in the paper and the responds to the reviewer’s comments are as flowing:

Reviewer #1:

1) Most of experiments were assayed with high dose validamycin (2 or 4 ug/insect). Please show any references on what was the basis for assay at these concentrations.

Response: The injection of high dose validamycin is based on previous research. - “Tang, B.; Yang, M.; Shen, Q.; Xu, Y.; Wang, H.; Wang, S. Suppressing the activity of trehalase with validamycin disrupts the trehalose and chitin biosynthesis pathways in the rice brown planthopper, Nilaparvata lugens. Pestic Biochem Physiol. 2017, 137, 81-90.”

2) Line 51, “thinning of the pupal epidermis,” is this sentence correct? In ref18, it is shown that thinning of the pupal epicuticle, not epidermis.

Response: Already revised in the manuscript.

3) Line 158, “at 35 C”, it lacks “°”.

Response: Already revised in the manuscript.

5) To clarify the effect of validamycin according to the duration of exposure, it is better that the rate of mortality and deformity are displayed on each day.

Response: During the statistical analysis, it was observed that the larvae injected with validamycin exhibited inactivity but still showed signs of life, making it difficult to define their state as dead. Therefore, only the mortality rate and deformity rate throughout the recorded period were documented. Almost all the inactive larvae with deformities eventually died.

6) Please mention more detail of dead larvae which were injected validamycin. Is that caused by defect of pupal cuticle or lack of energy.

Response: The larvae injected with the effective validamycin died with their bodies an orange color, while the control group showed no change. The possible causes of death could be due to defects pupal cuticle and a decrease in the activity of trehalase.

7) In figure 6, the malformations were detected at pupation. Do you think the similar effect are observed on larval ecdysis if validamycin is injected at 2nd larval instar?

Response: If validamycin is injected at 2nd larval instar, it may have similar effects on larval molting, due to the small size of the 2nd larval instar, the injection can easily cause massive mortality of the larvae.

Reviewer 2 Report

The authors studied the biochemical effects of injecting validamycin into oriental fruit fly larvae. This research may have value in the field of insect biochemistry, as it explores the action of validamycin on the trehalose pathway and could contribute to understanding the biology of this insect. It might also serve as a starting point for developing new pesticides. However, the current study is too far from practical field applications to justify its relevance. Growers will not inject a pesticide as a pest management approach, and the route of exposure influences biological outcomes. To be relevant one needs assays that expose pests to surface contamination, direct contact, or vapors as appropriate.

I acknowledge that there are many published papers justifying this sort of work based on a hypothetical link to pest management. However, the rationale is weak at best because the research is too distant from a field application. Most growers will not care about specific biochemical effects beyond a general mode of action as reported in IRAC. It also does not appear to address any of the issues relevant to a regulatory review board, at least not in the USA. That said, it is a registered fungicide and you touch on these issues in the very last paragraph of the discussion. I just see no need to start down this path.

Line 41) Replace the comma with a period, delete “while,” and make the remaining part a complete sentence.

Line 43) I am not sure how these two sentences relate to one another. At least initially I was expecting the five pathways (line 41) to be listed. You list one for trehalose synthesis and then pivot to trehalose degradation (hydrolysis) with nothing showing how these are related.

Line 43) Describe the five pathways in sufficient detail so that the rest of the paper makes sense.

Line 46) Within the context of this paper, is the distinction between cuticle and peritrophic lining relevant?

Line 48) Where does sfCHS fit in? The acronym is not defined. Is sfCHS present in other insects, or just S. frugiperda?

Line 48) include (Order: Family) at first use of scientific names.

Line 49) caused a high

Line 49) How does “additionally” combine sfCHS in S. frugiperda with Cht10 in A. albopictus?

Line 50) What is Cht10? Is it relevant to the oriental fruit fly? If not, why include it here?

Line 59) What is “Streptomyces hygroscopicus 5008?” I am not clear on the significance of the number.

Line 71) Is validamycin systemic so that it can penetrate a fruit and kill the larvae feeding there? Is killing larvae even relevant if the main damage is done at egg laying?

Line 74) Do not include results in the introduction.

Line 79) The introduction needs work. It has not properly prepared readers with sufficient background information.

1)      What are the pathways for trehalose synthesis? How does this feed into chitin biosynthesis? What are the enzymes as relevant to this paper?

2)      Where and how is validamycin used commercially? Target crops, target pests, application methods, and related information. If available, focus on crops where an application for oriental fruit fly control would be beneficial. As appropriate, acknowledge that the current label does or does not include this use: remember that the label is the Law. This section can also reassure readers that the product might be approved if it proves useful for insect management.

3)      What insects are proposed targets? I can find rice brown planthopper, Asian citrus psyllid, fall armyworm, common cutworm, sweet potato whitefly, and others. This background information may help convince readers that validamycin might be a useful insecticide, or at least worth further exploration. In part that depends on if some of the other papers can address use and efficacy questions. You might also briefly introduce where validamycin would be placed in the IRAC insecticide classification and how that might influence mode of action rotation schedules for resistance management.

a.       Has there been progress in turning validamycin into an insecticide, or is this just a rehashed excuse for doing awesome research?

Line 82) Insufficient. What did the larvae eat? Were adults fed?

Line 95) Is this 200 total or for each replicate within each treatment? What defines a treatment? If you have 200 total insects then that is 100 for 24H and 100 for 48H. Five are lost for Trehalose, five for glycogen, and five for glucose (=15) for each of 24 and 48H. That leaves no more than 60 for qRT-PCR at 48H. There are three rates of validamycin (0, 5, 10), so 20 each.

                Add a table, or some more text to make clear the number of replicates for each treatment. You have two time points (24, 48 H), and three doses (0, 5, 10) equals six treatments. To measure trehalose, glycogen, and glucose, then I will need 6 * 15 = 90 insects (section 2.4). Ten individuals were tested for chitin content where 6*10=60 insects (section 2.5). The remaining 50 individuals do not fit evenly into “four biological and 3 technical replicates for each gene” as neither 3, 4, nor 12 are even divisors of 50. Please correct the presentation so that I do not make whatever mistake I made above.

Line 102) remove the dash in supernatant.

Line 110) Were individuals analyzed individually or grouped into a single sample? My guess is pooled, as there are three replicates (30 insects in total).

Line 212) The graphs open the possibility of other interpretations. For both BdTPS and BdTPPB low doses were stimulatory but not significantly so for BdTPPB. high doses were significantly inhibited relative to low doses, but not different than the untreated control.

Line 244) very strong relative to what?

Line 248) efficient in what way? What does efficiency mean in this case?

Line 248) In what way is it environmentally friendly?

Line 250) studies (you have many citations)

Line 253) How would the application of validamycin as an insecticide influence the resistance management programs for using validamycin as a fungicide, or would these be in different environments?

Line 284) sentence structure. I am not sure what you are trying to say here.

Line 288) You would have to measure that in other insects. I do not follow the logic in this claim. At best it might indicate more validamycin targets in closely related species. However, that brings questions like how close is closely. Is this Diptera versus every other order, or just family, just genus, or subgroups within the genus?

Line 290) has potential

Line 297) It is bad to test against larvae and then suggest application to adults. Does validamycin have much of an effect on adult?

Line 297) can validamycin be systemic in the plant and thereby treat larvae?

Line 297) If only adults can be treated (and assuming efficacy) would such an application be useful in the field? The answer depends on where damage occurs, and if validamycin influences egg production, and how quickly it acts if it does influence egg production.

Line 301) Ok, so do a simple test with adults. Even just kill and count to show that validamycin can influence adult survival or egg production.

Line 305) These findings do not indicate that validamycin can be used as an insecticide. A theoretical insecticide does not protect crops, and there is a very long list of chemistries that can kill insects.

It is useful to discuss more of the history of validamycin and its agricultural use in the introduction. That paragraph can end with a statement indicating that this use is why you chose validamycin for this project. Stop there and do not continue suggesting that this research supports the use of validamycin for pest management because it does not. At best it could be used later when trying to identify a mode of action for resistance management purposes.

good, no real issues. 

Author Response

We are thankful to the referees and the Editor for pointing out some important modifications needed in our manuscript entitled “Validamycin Inhibits the Synthesis and Metabolism of Trehalose and Chitin in the Oriental Fruit Fly, Bactrocera dorsalis (Hendel)” (ID: insects-2498817). We have thoughtfully taken into account these comments. The explanation of what we have changed in response to the reviewers’ concerns is given point by point in the following pages. Revised portion are marked in red in the paper. The main corrections in the paper and the responds to the reviewer’s comments are as flowing:

Reviewer #2:

Line 41) Replace the comma with a period, delete “while,” and make the remaining part a complete sentence.

Response: Already revised in the manuscript.

Line 43) I am not sure how these two sentences relate to one another. At least initially I was expecting the five pathways (line 41) to be listed. You list one for trehalose synthesis and then pivot to trehalose degradation (hydrolysis) with nothing showing how these are related.

Response: Already revised in the manuscript.

Line 43) Describe the five pathways in sufficient detail so that the rest of the paper makes sense.

Response: Already revised in the manuscript.

Line 46) Within the context of this paper, is the distinction between cuticle and peritrophic lining relevant?

Response: The cuticle is the outermost layer of the epidermis, mainly composed of dead cells without nuclei. The peritrophic membrane is a complex composed of chitin, proteins, and polysaccharides, serving as the first natural barrier in insects to resist exogenous substances.

Line 48) Where does sfCHS fit in? The acronym is not defined. Is sfCHS present in other insects, or just S. frugiperda?

Response: sfCHS is chitin synthase of Spodoptera frugiperda. sfCHS present in S. frugiperda, CHS is present in other insects.

Line 48) include (Order: Family) at first use of scientific names.

Response: Already revised in the manuscript.

Line 49) caused a high

Response: Already revised in the manuscript.

Line 49) How does “additionally” combine sfCHS in S. frugiperda with Cht10 in A. albopictus?

Response: Chitin synthase and chitinase are key enzymes in the chitin synthesis pathway and degradation pathway.

Line 50) What is Cht10? Is it relevant to the oriental fruit fly? If not, why include it here?

Response: Cht10 is one of the chitinase enzymes, and a homologous Cht10 is also present in the oriental fruit fly.

Line 59) What is “Streptomyces hygroscopicus 5008?” I am not clear on the significance of the number.

Response: Already revised in the manuscript.

Line 71) Is validamycin systemic so that it can penetrate a fruit and kill the larvae feeding there? Is killing larvae even relevant if the main damage is done at egg laying?

Response: Currently, there have been no reports indicating that validamycin can penetrate the fruit and kill larvae feeding inside. We first study the toxicology and mechanism of action of the larvae and then proceed to study the adults.

Line 74) Do not include results in the introduction.

Response: Already revised in the manuscript.

Line 79) The introduction needs work. It has not properly prepared readers with sufficient background information.

Response: Already revised in the manuscript.

1) What are the pathways for trehalose synthesis? How does this feed into chitin biosynthesis? What are the enzymes as relevant to this paper?

Response: In insects, trehalose is primarily synthesized in the TPS/TPP pathway. The trehalose pathway is located at the beginning of the chitin synthesis pathway. TPS synthesizes trehalose through this pathway, while TRE breaks down trehalose to provide raw materials for chitin synthesis.  The enzymes associated with this article are as follows: Tre, HK, G6PI, GFAT, GPNA, PAGM, UNAP, CHS, Cht, IDGF, TPS, TPPs.

2) Where and how is validamycin used commercially? Target crops, target pests, application methods, and related information. If available, focus on crops where an application for oriental fruit fly control would be beneficial. As appropriate, acknowledge that the current label does or does not include this use: remember that the label is the Law. This section can also reassure readers that the product might be approved if it proves useful for insect management.

Response: validamycin (CAS: 37248-47-8) is an antibiotic produced by actinomycetes. It has strong systemic properties and is mainly used for controlling rice sheath blight disease. It is non-toxic and harmless to rice and even safe for humans and animals, making it an ideal environmentally friendly pesticide. It can also be used to prevent and treat diseases such as rice blast, corn spot disease, as well as vegetable, cotton, and bean crop diseases.

3) What insects are proposed targets? I can find rice brown planthopper, Asian citrus psyllid, fall armyworm, common cutworm, sweet potato whitefly, and others. This background information may help convince readers that validamycin might be a useful insecticide, or at least worth further exploration. In part that depends on if some of the other papers can address use and efficacy questions. You might also briefly introduce where validamycin would be placed in the IRAC insecticide classification and how that might influence mode of action rotation schedules for resistance management.

Response: Currently, validamycin is a confirmed fungicide primarily used for controlling rice sheath blight. It can also be used to treat rice bakanae disease, corn northern leaf blight, as well as diseases in vegetables, cotton, beans, and other crops. Since trehalose is widely present and highly important in organisms, we conducted toxicity tests using the larvae of B. dorsalis to verify its toxicological effects. These findings serve as a foundation for further research on adult insects and field application, and numerous results indicate the potential of validamycin as a new insecticide.

a. Has there been progress in turning validamycin into an insecticide, or is this just a rehashed excuse for doing awesome research?

Response: Validamycin is a microbial-based fungicide, and there have been numerous studies on its effects on insects, such as Nilaparvata lugens, Spodoptera litura, Spodoptera frugiperda, Bemisia tabaci, Diaphorina citri and Aedes aegypti. These results indicate that validamycin has a certain impact on the physiological metabolism and growth and development of insects, making it a type of novel insecticide that holds potential for further development and exploration.

Line 82) Insufficient. What did the larvae eat? Were adults fed?

Response: revised in the manuscript.

Line 95) Is this 200 total or for each replicate within each treatment? What defines a treatment? If you have 200 total insects then that is 100 for 24H and 100 for 48H. Five are lost for Trehalose, five for glycogen, and five for glucose (=15) for each of 24 and 48H. That leaves no more than 60 for qRT-PCR at 48H. There are three rates of validamycin (0, 5, 10), so 20 each.

Add a table, or some more text to make clear the number of replicates for each treatment. You have two time points (24, 48 H), and three doses (0, 5, 10) equals six treatments. To measure trehalose, glycogen, and glucose, then I will need 6 * 15 = 90 insects (section 2.4). Ten individuals were tested for chitin content where 6*10=60 insects (section 2.5). The remaining 50 individuals do not fit evenly into “four biological and 3 technical replicates for each gene” as neither 3, 4, nor 12 are even divisors of 50. Please correct the presentation so that I do not make whatever mistake I made above.

Response: Each treatment group was injected with 200 larvae. After injection the dead larvae in each treatment were daily removed and the remaining individuals were used for subsequent experiments.

Line 102) remove the dash in supernatant.

Response: Already revised in the manuscript.

Line 110) Were individuals analyzed individually or grouped into a single sample? My guess is pooled, as there are three replicates (30 insects in total).

Response: Yes, combine them into one sample for analysis. When measuring sugar content, a certain amount is required for extraction and analysis.

Line 212) The graphs open the possibility of other interpretations. For both BdTPS and BdTPPB low doses were stimulatory but not significantly so for BdTPPB. high doses were significantly inhibited relative to low doses, but not different than the untreated control.

Response: After injection of 5 μg/μL and 10 μg/μL of validamycin, both the glucose content and the expression levels of genes related to chitin synthesis pathway were significantly downregulated. However, for BdTPS and BdTPPB, low dose (5 μg/μL) had a stimulating effect, possibly due to a decrease in downstream content, which triggered a feedback mechanism in the insect body, leading to an increase in the amount of enzymes related to trehalose synthesis and thus producing more trehalose for downstream use. When injected at a high concentration, all aspects within the insect body were inhibited, resulting in no significant differences compared to the untreated group. In previous studies, similar results were observed, where the expression level of injecting 0.1 μg/μL of validamycin was significantly higher compared to injecting 10 μg/μL (Tang et al. 2017).

Line 244) very strong relative to what?

Response: In the referenced, validamycin A (VA) exhibits the strongest inhibitory effect on termite trehalase compared to validoxylamine A (VAA), validoxylamine B (VBB) and validamycin B (VB).

Line 248) efficient in what way? What does efficiency mean in this case?

Response: Research has shown that validamycin exhibits good efficacy against wheat and rice sheath blight. Here, "efficacy" refers to the ability of validamycin to significantly inhibit the sugar metabolism and chitin synthesis of the common cutworm.

Line 248) In what way is it environmentally friendly?

Response: It has been confirmed that validamycin is effective in preventing and controlling rice blast disease through extensive large-scale field experiments. It is non-toxic and harmless to rice plants, and even highly safe for humans and animals.

Line 250) studies (you have many citations)

Response: Already revised in the manuscript.

Line 253) How would the application of validamycin as an insecticide influence the resistance management programs for using validamycin as a fungicide, or would these be in different environments?

Response: Validamycin, as a fungicide, is primarily used to control rice sheath blight. It can also be used to manage rice blast, maize northern leaf blight, as well as diseases in vegetables, cotton, and leguminous crops. However, if used as an insecticide, we typically apply it in orchards. Orchards rarely use fungicides like validamycin, and there is no temporal and spatial overlap, resulting in a very low possibility of developing resistance.

Line 284) sentence structure. I am not sure what you are trying to say here.

Response: Already revised in the manuscript.

Line 288) You would have to measure that in other insects. I do not follow the logic in this claim. At best it might indicate more validamycin targets in closely related species. However, that brings questions like how close is closely. Is this Diptera versus every other order, or just family, just genus, or subgroups within the genus?

Response: We have only found TPP in Diptera, and our research has revealed that validamycin can inhibit TPP. Therefore, we speculate that there may be more potential targets in Diptera, and further validation can be conducted in the future.

Line 290) has potential

Response: revised in the manuscript.

Line 297) It is bad to test against larvae and then suggest application to adults. Does validamycin have much of an effect on adult?

Response: The preliminary investigation indicated that validamycin treatment on adults may affect their fecundity, and the thorough evaluation is still ongoing.

Line 297) can validamycin be systemic in the plant and thereby treat larvae?

Response: Validamycin is a systemic fungicide, and currently, there are no reports indicating that it can be used to treat larvae in plants.

Line 297) If only adults can be treated (and assuming efficacy) would such an application be useful in the field? The answer depends on where damage occurs, and if validamycin influences egg production, and how quickly it acts if it does influence egg production.

Response: If the treatment with validamycin has an effect on adults, there will undoubtedly be practical effects for its field application. However, further research is needed to determine its effectiveness.

Line 301) Ok, so do a simple test with adults. Even just kill and count to show that validamycin can influence adult survival or egg production.

Response: Preliminary experiments were conducted by injecting validamycin into adult insects, and compared to the control group, the number of egg-laying of adults injected with a high concentration was significantly lower than that of the control group. However, this result requires further validation.

Line 305) These findings do not indicate that validamycin can be used as an insecticide. A theoretical insecticide does not protect crops, and there is a very long list of chemistries that can kill insects.

Response: These findings indicate the insecticidal potential of validamycin against dipteran insects, as well as its potential as a novel insecticide for pest control purposes. Subsequent evaluation to be studied

It is useful to discuss more of the history of validamycin and its agricultural use in the introduction. That paragraph can end with a statement indicating that this use is why you chose validamycin for this project. Stop there and do not continue suggesting that this research supports the use of validamycin for pest management because it does not. At best it could be used later when trying to identify a mode of action for resistance management purposes.

Response: revised in the manuscript.